# Acute Localized Exanthematous Pustulosis (ALEP) Caused by Topical Application of Minoxidil

**DOI:** 10.3390/jcm12030831

**Published:** 2023-01-20

**Authors:** Michael Makris, Antonios Kanelleas, Niki Papapostolou, Maria Pisimisi, Alexander C. Katoulis

**Affiliations:** 1Allergy Unit “D.Kalogeromitros”, 2nd Department of Dermatology and Venereology, Medical School, National and Kapodistrian University of Athens, University General Hospital “Attikon”,12462 Athens, Greece; 22nd Department of Dermatology and Venereology, Medical School, National and Kapodistrian University of Athens, University General Hospital “Attikon”, 12462 Athens, Greece; 3Dermathens Private Practice, 10675 Athens, Greece

**Keywords:** Acute Localized Exanthematous Pustulosis (ALEP), minoxidil, Acute Generalized Exanthematous Pustulosis (AGEP), cutaneous drug reaction, patch tests, hypersensitivity reaction, pustules

## Abstract

Acute Localized Exanthematous Pustulosis (ALEP) is a rare skin reaction characterized by the sudden onset of multiple, small, sterile, non-follicular pustules in an erythematous and edematous base succeeding systemic drug administration. ALEP is considered a subtype of Acute Generalized Exanthematous Pustulosis (AGEP), although the exact pathogenic mechanism of the disease remains poorly defined. Numerous drugs have been implicated in the pathogenesis of ALEP, while contact mechanisms have also been reported. Herein, we describe the first case of ALEP attributed to minoxidil in a female patient with androgenetic alopecia. The positivity of patch tests and the topical application of minoxidil proposes a contact-induced hypersensitivity reaction. Identifying new agents—including minoxidil—which serve as inducers of drug-specific T-cell-mediated responses in the clinical spectrum of ALEP, adds further value in understanding the complex, yet unknown, pathophysiological mechanisms of this rare drug hypersensitivity reaction.

## 1. Introduction

Acute Localized Exanthematous Pustulosis (ALEP) is a rare, localized variation of Acute Generalized Exanthematous Pustulosis (AGEP), characterized by the unanticipated onset of numerous small, sterile, non-follicular, pinhead-sized pustules in an erythematous base, commonly succeeding systemic drug administration [1,2]. Still, a contact mechanism has been reported [3,4]. Almost 40 cases of ALEP have been described in the literature in both paediatric and adult populations since the description of the first case in 2005 by Prange et al. [1,5]. However, data on pathogenic mechanisms, clinical courses, and diagnostic procedures and treatment remain scarce. Herein, we describe a case of a female patient presenting with a cutaneous drug reaction consistent with ALEP following topical administration of a compounded formulation of 17-a-estradiol 0.05% and minoxidil 5% in Trichofoam^TM^ for androgenetic alopecia.

## 2. Case Description

A 31-year-old Caucasian female with a known history of androgenetic alopecia presented to the Emergency Department of the 2nd Dpt of Dermatology and Venereology Clinic in ‘Attikon’ University General Hospital, Athens, Greece due to a sudden outbreak of multiple pruritic pustules on the frontal scalp, forehead, and neck. The eruptions appeared 48 h after the application of a compounded foam formulation containing 17-a-estradiol 0.05% and minoxidil 5% in Trichofoam^TM^ which acted as an excipient (Fagron Derma Pack MET-A). This was prescribed by her dermatologist and, according to his instructions, was applied once daily on the scalp.

The patient was diagnosed with androgenetic alopecia fifteen years ago and has since been on intermittent treatment for varying time periods with different minoxidil formulations with good therapeutic responses. However, approximately four years after the initial onset of treatment, the third or fourth day of topical application of an unknown minoxidil formulation prescribed by her dermatologist once daily, she developed a mild, pustular-free, pruritic eruption on the scalp succeeding the topical application that fully resolved without treatment within four days after the cessation of minoxidil application. Since then, treatment courses with minoxidil ceased until her recent presentation in the emergency department. Apart from that, the patient had no personal or family history of AGEP and/or other atopic and allergic conditions.

At her visit to the Emergency Department, the physical examination revealed multiple non-follicular pustules affecting the scalp, forehead, and neck on an erythematous, edematous background. (Figure 1 and Figure 2) In addition, enlarged cervical lymph nodes were noted, with the patient being febrile (temperature of 37.8 °C) and lacking mucosal or joint involvement. The complete blood count was normal. Erythrocyte sedimentation rate, C- reactive protein, liver and renal function were within normal limits. Bacterial and fungal blood cultures were negative while Herpes Simplex Virus serology was also negative.

According to our instructions, the minoxidil-estradiol formulation was immediately discontinued, and the patient was treated with potent topical steroids. Although a punch skin biopsy was recommended, the patient denied it. The resolution of the pustular lesions occurred within seven days of the withdrawal of the minoxidil-estradiol formulation, while post-pustular desquamation was observed until full skin recovery approximately 20 days later.

The diagnostic work-up with patch tests was performed on all applied agents (Minoxidil 5%, 17-a-estradiol 0.05%, Trichofoam^TM^ FAGRON) 8 weeks after the resolution of the eruption. The 48 h and 72 h readings revealed: a. positive reaction to: a. minoxidil diluted in Trichofoam^TM^ and b. minoxidil and estradiol diluted in Trichofoam^TM^. Negative results were noted to a. Trichofoam^TM^ on its own, b. estradiol diluted in Trichofoam^TM^, c. estradiol diluted in ethoxyl (vehicle solution) and d. ethoxyl on its own. (Figure 3) Thus, minoxidil hypersensitivity per se was proven.

Based on the clinical presentation of the eruption with the characteristic localized morphology of multiple non-follicular pustules, along with the clear temporal association with minoxidil application and the resolution following the drug withdrawal, the diagnosis of minoxidil-induced ALEP was reached. The positivity of the patch tests and the negativity of bacterial and fungal cultures further supported the clinical diagnosis, which agreed with the EuroSCAR criteria for AGEP and the recently proposed diagnostic criteria for ALEP [6,7,8].

## 3. Discussion

To our knowledge, this is the first reported case of minoxidil-induced ALEP. Furthermore, the topical application of minoxidil clearly shows a contact-induced hypersensitivity reaction. Most cases of ALEP described in the literature are attributed to systemically administrated drugs ranging from antibiotics [9,10,11,12,13,14] to a variety of other drug categories (NSAIDS, platinum agents, antiepileptic drugs, anticoagulant agents, monoclonal antibodies, 5-a reductase inhibitors, and antineoplastic drugs) as well as herbal products [2,15,16,17]. However, contact mechanisms have been described previously [3,4].

Minoxidil is a hypertrichotic agent generally used to treat androgenetic alopecia and it was initially approved in 1988 by the Food and Drug Administration (FDA) for the treatment of men with mild to moderate androgenetic alopecia [18]. An additional FDA approval followed in 1992 regarding the treatment of female pattern hair loss [19]. Topical minoxidil 5% solution is generally well tolerated, although adverse effects like pruritus and contact dermatitis have been reported [20]. Despite many cases of contact dermatitis having been reported in the literature, pustular contact dermatitis due to minoxidil is rarely described [21]. The differential diagnosis of acute contact dermatitis from ALEP is difficult. However, the presence of non-follicular pustules, the absence of follicular vesicles and mild pruritus, are well-described characteristics of ALEP that assist its diagnosis of both pustular and non-pustular contact dermatitis [1]. Moreover, in our case, the presence of cervical lymphadenopathy, fever, and the rapid resolution of skin lesions with subsequent desquamation, following the drug withdrawal, favour ALEP diagnosis.

Although skin biopsy is always a valuable diagnostic tool and was recommended in our case, it is not considered essential for the diagnosis and, accordingly, is not included in the recently proposed diagnostic criteria for ALEP [7].

Furthermore, most cases of eruptions attributed to minoxidil formulations are thought to be due to minoxidil solution vehicle rather than minoxidil per se [20,22]. In our case, in vivo patch tests revealed “sensitization” or drug-specific T-cells to minoxidil per se acquitting vehicles and estradiol. This is further supported by the patient’s previous pruritic eruption four years ago following the topical application of minoxidil. In accordance, although this previous skin rash is not consistent with ALEP it supports the existence of minoxidil-specific T-cell clones which were initially triggered four years ago and then, as the patient was not treated with minoxidil, they remained inactivated, and the patient experienced no rash. However, 48 h after the first subsequent administration, even four years later, specific memory T-cells were recruited in the skin resulting in ALEP presentation. Whether the previous eruption could result in ALEP if the drug was not discontinued on time remains an unanswered question.

As ALEP has been characterized as a subtype of AGEP, the proposed underlying mechanisms are mainly derived from the extrapolation of our knowledge of AGEP [23,24]. Both forms of Exanthematous Pustulosis are thought to be mediated by drug-specific T-cells (IVd reaction according to Gells and Coombs modified classification) [25,26,27]. In line with this theory, positive patch tests and lymphocyte transformation tests can be shown in about 50% of the patients, although the sensitivity and specificity of diagnostic measures show great variations [28,29,30]. The Th17 cells are thought to contribute to neutrophils infiltration in skin pustules enhancing the production of interleukin-8 (IL-8) leading to chemotaxis of neutrophils with subsequent formation of sterile pus, transforming vesicles to pustules [31,32].

In our case, the positivity of patch tests to minoxidil per se enables a glimpse into the unknown pathogenic mechanism of this new offending agent-induced ALEP, suggesting the involvement of minoxidil-specific T-cells (CD8+ and CD4+). In particular, after the topical application of minoxidil, the antigen-presenting cells in the skin intake, process, and present minoxidil to T-cells using the major histocompatibility complex (MHC). Firstly, activated CD8+ cells induce keratinocytes apoptosis and formation of vehicles, and then, both specific to minoxidil, CD8+ and CD4+ cells secrete large amounts of interleukin 8 (IL-8) which acts as neutrophilic accumulation cytokine leading to chemotaxis of neutrophils and the formation of sterile pustules. Although the proposed mechanism is far from being elucidated, the localization of the eruption strongly suggests that the drug acts as hapten binding basal keratinocytes [7].

## 4. Conclusions

To date, minoxidil, one of the few licenced drugs for female pattern hair loss, and thus frequently used in dermatology, has never been suspected as a causative drug for ALEP. Despite being reported as a cause of allergic contact dermatitis, with anecdotal referrals of pustular variations, its implementation in ALEP pathogenesis is proposed for the first time. The identification of new agents—including minoxidil—that serve as inducers of drug-specific T-cell-mediated responses in the clinical spectrum of acute localized pustulosis adds further value in understanding the complex, yet unknown, pathophysiological mechanisms of this rare drug hypersensitivity reaction.

## Figures and Tables

**Figure 1 jcm-12-00831-f001:**
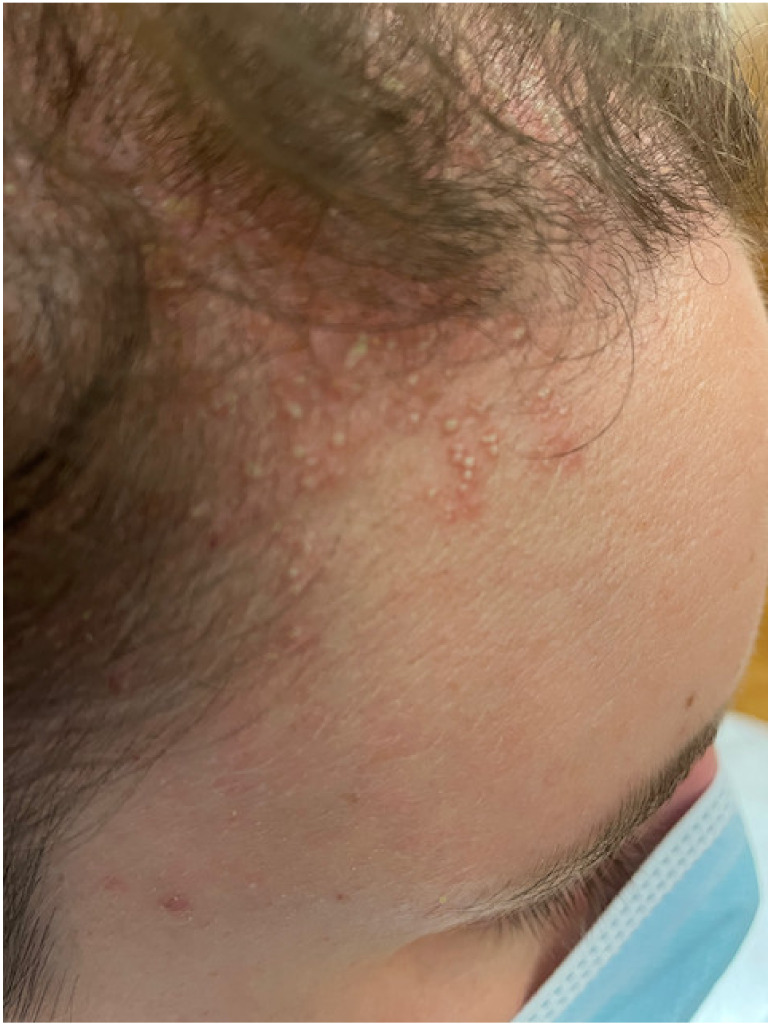
Presence of multiple non-follicular pustules on an erythematous base distributed on the forehead.

**Figure 2 jcm-12-00831-f002:**
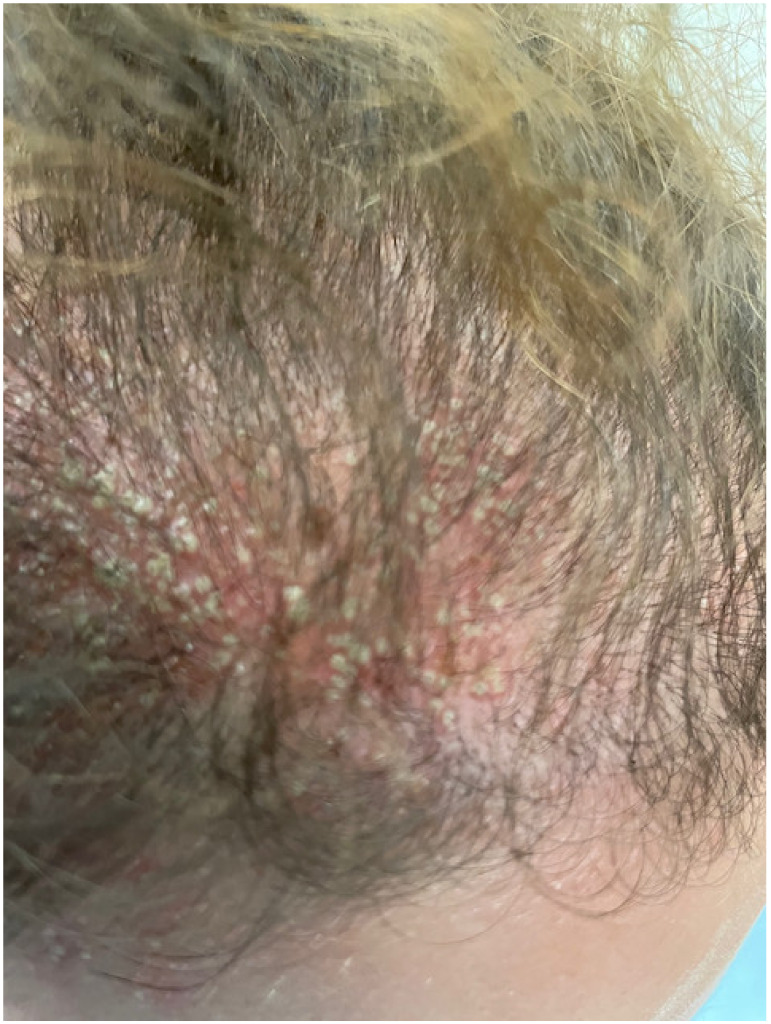
Presence of multiple non-follicular pustules on an erythematous base distributed on the scalp.

**Figure 3 jcm-12-00831-f003:**
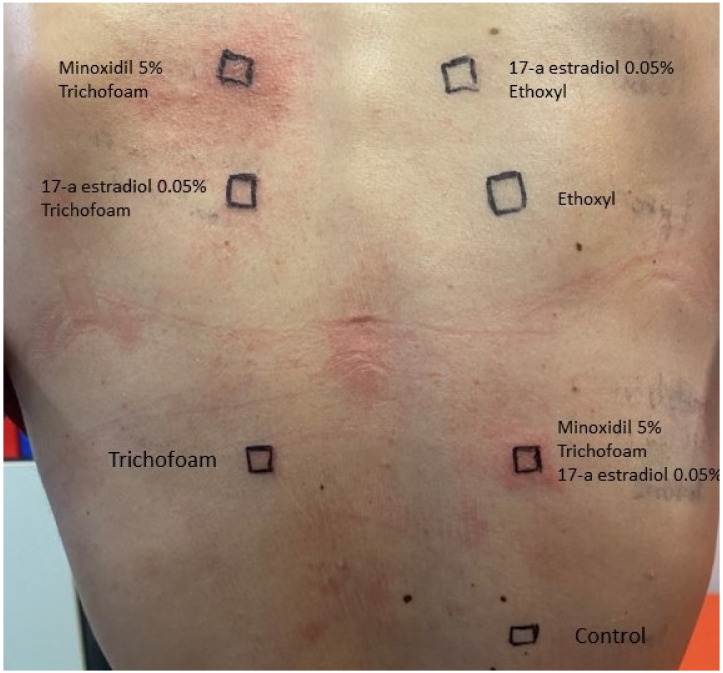
Positive patch tests to a. minoxidil 5% diluted in Trichofoam^TM^ and b. minoxidil 5% and 17-a estradiol 0.05% diluted in Trichofoam^TM^ and negative results to a. Trichofoam^TM^, b. 17-a estradiol diluted in Trichofoam^TM^, c. 17-a estradiol diluted in ethoxyl, and d. ethoxyl per se (vehicle solution).

## Data Availability

Not applicable.

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
