# Peer review of "Acute Localized Exanthematous Pustulosis (ALEP) Caused by Topical Application of Minoxidil"

_jcm, 2023, doi:10.3390/jcm12030831_

Round 1

Reviewer 1 Report

The paper entitled "Acute Localized Exanthematous Pustulosis (ALEP) caused by topical application of minoxidil" is highly interesting and in scope of the journal.

1. It is necessary to add a review of possible pathogenetic mechanisms of minoxidil-induced ALEP development in the Discussion section.

2. Rewrite the conclusion! Conclusions (lines 134-141) are exactly the same as the last paragraph of the Discussion section (lines 124-131). The conclusion section should be based on an analysis of our own results, taking into account the literature data discussed in the Discussion section.

Reviewer 2 Report

This is a well prepared and important addition to the literature. The authors just need to correct a few minor issues.

1.     Line 40 patient “presented” with a cutaneous drug reaction consistent with ALEP …..

Rather presenting with OR who presented with ….

2.     Line 41 and 48 “compound formulation” rather “compounded formulation”

3.     Line 61 superscript temperature.

4.     Line 61 ‘and lacking mucus….” I think the authors were referring to ‘mucosal’ not mucus

5.     Line 97 “Despite many cases of contact dermatitis have been reported in the literature, pustular contact dermatitis due to minoxidil is rarely described. Rather “Despite many cases of contact dermatitis having been reported in the literature, pustular contact dermatitis due to minoxidil is rarely described”

6.     Line 100” Besides, acute contact dermatitis is rather unchallenging to differentiate from ALEP as the presence of non-follicular pustules in the absence of follicular vesicles along with the rather limited pruritus are well defined characteristics of ALEP and support the diagnosis over contact dermatitis -pustular or not.” Please simplify this sentence.

Reviewer 3 Report

1. Your report did not describe the patient's history of allergies and history of AGEP, which is very important.

2.”Discussion” mentions the patient's pruritic eruption four years ago following the topical application of minoxidil. You should describe the condition at the time in detail.

3. What was the brand and daily dosage of Minoxidil the patient used? Why ALEP appeared for the first time in four years? Whether the patient changed the commodity source or dosage of Minoxidil?

4. Were drug ingredients in the patch tests isolated from the compound formulation of minoxidil 5% and 17-a-estradiol 0.05% the patient used, or from other brand drugs?

5. ”Conclusions” repeated the content of ”Discussion” completely and should be refined.

Reviewer 4 Report

The authors reported an interesting case of ALEP coused by minoxidil. The comments are followed:

1. Please keep the ingredients consistent in Line 41 and Line 48;

2. Please change the temprature in Line 61: 37.8ο C into 37.8°C;

3. In the discussion section, I suggest a more concise and orderly discussion.

  •  
